# Dual Role of B Cells in Multiple Sclerosis

**DOI:** 10.3390/ijms24032336

**Published:** 2023-01-25

**Authors:** Gaurav Kumar, Robert C. Axtell

**Affiliations:** Arthritis and Clinical Immunology Research Program, Oklahoma Medical Research Foundation, Oklahoma City, OK 73104, USA

**Keywords:** multiple sclerosis, B cells, disease-modifying therapies

## Abstract

B cells have emerged as an important immune cell type that can be targeted for therapy in multiple sclerosis (MS). Depleting B cells with anti-CD20 antibodies is effective in treating MS. Yet, atacicept treatment, which blocks B-cell Activating Factor (BAFF) and A Proliferation-Inducing Ligand (APRIL), two cytokines important for B cell development and function, paradoxically increases disease activity in MS patients. The reason behind the failure of atacicept is not well understood. The stark differences in clinical outcomes with these therapies demonstrate that B cells have both inflammatory and anti-inflammatory functions in MS. In this review, we summarize the importance of B cells in MS and discuss the different B cell subsets that perform inflammatory and anti-inflammatory functions and how therapies modulate B cell functions in MS patients. Additionally, we discuss the potential anti-inflammatory functions of BAFF and APRIL on MS disease.

## 1. Introduction

Multiple sclerosis (MS) is a chronic immune-mediated disease of the central nervous system (CNS) characterized by demyelination and subsequent axonal damage resulting in the loss of motor and sensory functions [1]. MS is one of the most common causes of neurological disability, especially in young adults between 20 and 40 years of age [2]. According to the National Multiple Sclerosis Society (NMSS), more than 2.3 million people are affected by MS worldwide, and the incidence continues to increase. The burden of this complex disease is associated with the reduced quality of life and increased mortality of patients [3].

The discovery of oligoclonal bands (OCBs) in the CSF of patients with MS in 1960 [4] was the first indication that B cells could be an important cell type for the etiology of MS. It was not until 1999 that B cells were shown to play a major role in MS pathology using the mouse model, experimental autoimmune encephalomyelitis (EAE) [5]. These animal studies generated an interest in testing anti-CD20 antibody B cell depletion therapy for MS, which is now proven to be highly effective in reducing relapse rates and slowing disability in MS [6,7]. Around the same time as the discovery of a pathogenic B-cell function in MS, other research groups showed that B cells may also have anti-inflammatory functions in the EAE model [8,9]. Subsequently, the failure of the atacicept trial, which unfortunately exacerbated disease in patients, demonstrated that B cells can exert an anti-inflammatory effect in MS [10].

These clinical and experimental observations demonstrate that the role of B cells can have diametrically opposing functions in MS. In this review article, we explore the clinical and basic immunological research that has shed light on the highly nuanced function of B cells in MS.

## 2. Dual Role of B Cells in MS

The functional role of B cells in MS is highly nuanced, and there is much left to be understood. It is generally believed that B cells contribute to MS by producing autoantibodies, expressing inflammatory cytokines, and presenting antigens to T helper cells [5]. The anti-CD20 clinical trials provide solid evidence that B cells have an inflammatory function in MS [11]. However, clinical trials with atacicept blocking BAFF and APRIL, cytokines important in B cell survival and function, increased disease activity in MS, demonstrating that some B cell subsets have anti-inflammatory functions [10].

After exiting the bone marrow, B cells undergo a series of developmental stages to become mature B cells [12]. The first B cells to emerge from the bone marrow are the immature transitional B cells. This subset of B cells is in a transient developmental stage that is still undergoing antigen receptor selection. The transitional B cells that survive antigen receptor selection eventually develop into mature naïve B cells. In secondary lymphoid tissues, mature naïve B cells can encounter antigens and become activated and develop into germinal center B cells, class-switched (CS) memory B cells, and plasmablasts [12]. The B cell subsets along this developmental pathway have different functions; some have anti-inflammatory properties and others have inflammatory properties [13,14]. The different subsets of B cells that have been identified to have either inflammatory or anti-inflammatory function in MS are discussed below and summarized in Table 1.

### 2.1. Inflammatory Subsets of B Cells

Even though it is known that B cells contribute to the pathology of MS, it is unclear which B cell subsets and what effector function is most important for driving disease in patients. Class-switched memory B cells are a subset that have been found to be highly inflammatory in MS. It has been reported that patients experiencing an active relapse have higher numbers of CS memory B cells in the blood compared with patients in remission [6,7,11,15]. Memory B cells from MS patients exhibit increased expression of CD40 and HLA-DR, suggesting an increased capacity for antigen presentation by B cells to CD4 T cells in MS patients [16]. In mice with EAE, MHCII deletion specifically on B cells reduces the generation of myelin-specific TH1 and TH17 cells and blocks the development of clinical signs of EAE induced by recombinant human MOG. In another EAE mouse model, the expansion of antigen-specific B cells during CNS inflammation drives the cognate interaction between B cells and CD4 T cells and elicits neuro-inflammation in mice [17]. These studies suggest the importance of antigen-presenting functions of B cells in mediating neuro-inflammation and a direct B cell–T cell interaction in MS [18].

The potential to produce inflammatory cytokines is another mechanism by which memory B cells contribute to the pathogenesis and severity of MS. Memory B cell subsets from MS patients produce high levels of the TNF-α, IL-6, and GM-CSF cytokines compared with B cells from healthy controls, which can activate the inflammatory function of T and myeloid cells to contribute to the severity of neuro-inflammation [14,19]. In MS patients, the depletion of B cells led to decreased inflammatory response by cells, which was directly associated with depletion of GM-CSF-producing B cells [14,20]. EAE experiments demonstrated that B cells are major producers of IL-6, and mice with IL-6 deficiency in B cells have less severe disease than controls [19]. The lack of IL-6 in B cells decreases the differentiation of myelin-specific TH17 cells to reduce disease severity in mice.

Autoantibody-producing plasmablasts have also been implicated in driving the pathogenesis of MS. The presence of oligoclonal bands by clonally expanded B cells in the CSF of MS patients, which remains the hallmark for the diagnosis of MS, provides strong evidence for the functional role of antibody-secreting cells in the pathogenesis of MS [21]. Presumably, antibodies produced in the CNS of MS patients bind to either myelin or other nervous system proteins, which then activate the complement system to drive neuro-inflammation and CNS tissue damage. The search for antigens targeted by autoantibodies in MS is a longstanding research question. Recently, there has been a resurgence of interest in the link of MS and Epstein–Barr virus (EBV) infection [22]. The causal mechanisms of EBV in driving MS have not been fully established. However, there are multiple reports suggesting that antigenic mimicry between EBV proteins and CNS antigens might be involved with the etiology of MS. We recently found that serum antibodies from MS patients target EBV nuclear antigen 1 (EBNA-1) at amino acid residues 411–426 and that these antibodies cross-react with myelin basic protein [23]. Other researchers identified that antibodies from MS cross-react with the 411–440 amino acid stretch of EBNA-1 and anoctamin 2 (ANO2), a protein found in axons [24]. Another study also found that clonally expanded antibodies in the CSF of MS patients target EBNA-1 residues 386–405 and cross-react with GlialCAM, a CNS antigen [25]. Although these studies suggest different CNS-antigen targets for MS, they all identified that antibodies from MS bind to a contiguous region of EBNA-1, which is not observed in healthy volunteers. These data suggest that antibody responses to EBV may drive immune-mediated CNS pathology in MS.

### 2.2. Anti-Inflammatory Subsets of B Cells

B cells may also have anti-inflammatory effects that could help induce remissions in MS patients. The strongest evidence for an anti-inflammatory function of B cells in MS was shown in a phase II open-label clinical trial of atacicept. The study found that atacicept, a BAFF and APRIL blocker that reduces B cells in vivo, significantly increased relapse rates in MS (ClinicalTrials.gov: NCT00642902) [10]. This failure was surprising due to the success of rituximab in RRMS and anti-BAFF therapy in lupus; yet, it demonstrates the nuanced role of B cells in autoimmunity. There are now numerous reports of various B cell subsets that suppress disease activity in MS and other autoimmune diseases [26,27,28,29]. The immature transitional B cells (defined as CD19^+^, CD38^++^, and CD24^++^) have anti-inflammatory functions defined by the expression of IL-10 and the capability to inhibit T cell responses [26,28]. We, and others, have shown that drugs that skew the ratio of B cells toward the anti-inflammatory transitional B cell population are associated with a favorable response to therapy in MS patients [13,30,31]. Subsets of terminally differentiated B cells may also have anti-inflammatory effects in MS. Mature CD5^+^ B10 cells are potent producers of IL-10 and inhibit experimental autoimmune diseases in mice [32,33]. An analogous subset of human memory B cells that express the receptors TIM1, CD43, and CD27 may also have anti-inflammatory functions in autoimmune disease. These are innate-like B cells that produce high levels of low-affinity auto-reactive IgM [34] and produce IL-10 to regulate T cell activity [35]. More recently, it was suggested that IgA-producing plasmablasts have an anti-inflammatory function in MS and in EAE. The common trait between these subsets of anti-inflammatory B cells is their expression of IL-10, which is a potent inhibitor of T cell and myeloid cell activity. However, other cytokines have been implicated in the anti-inflammatory function of B cells, including the secretion of cytokines IL-35 and TGF-β and the TIGIT receptor, which redirect T cells toward a Foxp3 Treg phenotype and attenuate autoimmunity [36,37,38].

## 3. Therapies That Restrict Inflammatory B Cells and Augment Anti-Inflammatory B Cells in MS

With the success of different anti-CD20 therapies, research has focused on how therapies modulate inflammatory and anti-inflammatory B cell function in MS. Below, we discuss the current literature surrounding the effects of different MS therapies on B cells (summarized in Table 2).

### 3.1. Anti-CD20 Therapies

CD20 is expressed by the vast majority of B cells, from pre-B cells to memory B cells, excluding earlier stages (pro-B cells) and plasma cells. As the plasma cells are not eliminated, the levels of autoantibodies were shown to remain unchanged after anti-CD20 therapy [6]. Until now, three different anti-CD20 therapies have been used in MS, including rituximab, ocrelizumab, and ofatumumab. Rituximab administration was reported to successfully reduce the relapse rates as well as the formation of lesions in MS patients. Similarly, various clinical trials with ocrelizumab and ofatumumab have also shown improvement in disease severity [7,39]. These therapies are efficacious even though they fail to eliminate plasma cells, suggesting a critical role of antigen presentation and cytokine production as the major functions of B cells in the pathophysiology of MS rather than autoantibody production [40]. In mice, the administration of anti-CD20 before inducing EAE worsens the disease severity; in contrast, the administration of anti-CD20 therapy after clinical signs of EAE have developed ameliorates disease. These data suggest that anti-inflammatory B cells inhibit the induction of disease, but memory B cells propel disease activity later in the course of disease [33]. Another report demonstrated that anti-CD20-mediated depletion of IL-6-producing B cells led to reduced infiltration of inflammatory T cells into the CNS, reduced levels of IL-17 and IFN-γ, and subsequent disease improvement [19]. Altogether, therapies directly targeting B cells and their efficacy in MS patients suggest a dominant role played by B cells in the disease.

### 3.2. Interferons

Recombinant interferon beta, IFNβ-1a and IFNβ-1b, remains a widely used treatment for RRMS patients. IFN-β elevates BAFF and APRIL levels (two cytokines important for B cell and plasma-cell development [41]), B cell activity, and antibody production [42,43,44,45]. Our laboratory was the first to report that alterations in B cell subset profiles are associated with a good response to disease-modifying therapy (DMT) in RRMS patients. In a cross-sectional study that assessed B cell populations, we discovered that IFN-β treatment skews the ratio of B cells away from an inflammatory class-switched (CS) memory B cell phenotype toward an anti-inflammatory transitional B cell phenotype in MS patients [13]. In addition, we found that the transitional B cells from these IFN-β responder MS patients are potent producers of IL-10 [13]. The elevation in transitional B cells and decrease in CS memory B cells by IFN-β treatment have since been confirmed by other independent studies [31]. Furthermore, it was reported that patients experiencing an active relapse while on IFN-β treatment have lower numbers of transitional B cells and higher CS memory B cells in the blood [46]. Others showed that IFN-β therapy mediates its effect by specifically depleting the memory B cells through apoptotic cell death. It was speculated that increased BAFF levels due to IFN-β therapy binds to TACI, which is highly expressed on memory B cells, and mediates apoptotic cell death without affecting plasma cells in the bone marrow of RRMS patients [47].

### 3.3. Dimethyl Fumarate

Dimethyl fumarate (DMF) is a commonly prescribed and well-tolerated therapy for RRMS patients; various clinical trials have demonstrated its efficacy in reducing relapse rates and CNS inflammatory lesions [48,49]. DMF reduces memory B cells through apoptotic cell death, but spares naïve B cells, including IL-10-producing transitional B cells, in RRMS patients. Effective DMF treatment was also associated with decreased production of pro-inflammatory cytokines GM-CSF, IL-6, and TNF-α [50]. A similar study of DMF therapy reported an increase in IL-10-producing transitional B cells and B10 cells in RRMS patients [30]. Therefore, DMF may mediate its therapeutic effect by increasing the ratio between anti-inflammatory and inflammatory B cells in RRMS.

### 3.4. Fingolimod

Fingolimod was the first orally administered drug approved by the FDA for the treatment of RRMS. Fingolimod is structurally similar to sphingosine; therefore, it binds to the sphingosine receptors, causing internalization and degradation of the receptors. Sphingosine receptors are expressed on the surface of lymphocytes and aid them to egress from the secondary lymphoid tissues [51,52]. Fingolimod-treated MS patients experience a significant decrease in circulating lymphocytes, including B cells, and consequently have low numbers of inflammatory cells infiltrating into the CNS. Fingolimod therapy increases the circulation of naïve B cells and selectively reduces memory B cells. The therapy also induces the expression of co-stimulatory molecules CD80 and CD86 on B cells, which are thought to increase the activity of the remaining B cell pool [53]. In an interesting study, fingolimod treatment increased the anti-inflammatory IL-10-producing B cells, augmented their migratory capacity into the CNS, and thereby reduced disease severity [54].

The dual role of B cells in the pathophysiology of MS patients is evident from clinical studies with different therapies for MS. Anti-CD20 therapies, IFN-β, DMF, and fingolimod all reduce memory B cells while maintaining or increasing anti-inflammatory B cells and sparing plasma cells. In contrast, atacicept depletes transitional B cells and plasma cells but spares memory B cells and thereby worsens MS. The failure of atacicept in MS opened new avenues for understanding the complex functions of B cells. A rational approach to understanding the mechanism of action of atacicept would be to explore the signaling and functions of BAFF and APRIL and their receptors.

## 4. Anti-Inflammatory Role of BAFF and APRIL on B Cells

### 4.1. Blockade of BAFF and APRIL Exacerbates MS

Atacicept is a human recombinant fusion protein that binds to BAFF and APRIL, cytokines required for B cell differentiation and development, thereby stopping them from interacting with their receptors, BAFFR, TACI, and BCMA. Atacicept consists of the Fc region of human IgG1 linked to the extracellular binding domain of TACI. Atacicept treatment reduces serum immunoglobulin levels and the numbers of circulating B cells, mainly immature and naïve B cells, as well as plasma cells but not B cell progenitors or memory B cells. In the EAE mouse model, atacicept therapy significantly delayed the onset of disease, which was associated with reduced infiltration of B cells into the CNS and decreased circulating mature B cells and serum IgM and IgG levels [55]. Therefore, it was hypothesized that atacicept therapy could be effective in reducing the disease severity in MS patients, as it could limit the antibody-producing plasma cells. A randomized, placebo-controlled, phase 2 trial of atacicept in multiple sclerosis (ATAMS) was prematurely terminated because atacicept-treated patients had increased relapse rates and numbers of inflammatory lesions, even though the drug successfully reduced the number of circulating mature B cells and serum immunoglobulins. This study clearly indicates an anti-inflammatory role of B cells in MS [10]. Though the precise mechanism behind the failure of atacicept still remains elusive, several speculations could be made based on the available data. A logical mechanism is that atacicept preferentially eliminates anti-inflammatory B cell subsets that produce IL-10.

### 4.2. Effects of BAFF and APRIL Signaling in Neuro-Inflammation

Signals initiated by BAFF and APRIL are necessary for the survival and maintenance of B cells [56,57]. BAFF and APRIL bind to and signal through three receptors: BAFF receptor (BAFFR), transmembrane activator and calcium modulator and cyclophilin ligand interactor (TACI), and B cell maturation antigen (BCMA) [58,59] (Figure 1). BAFF binds to all three receptors, whereas APRIL binds only to TACI and BCMA. The expression of these receptors on B cells is dynamic, as their expression levels change at different stages of B cell development (Figure 2). BAFFR is expressed early in B cell development at the transitional B cell stage, and high levels of this receptor are maintained through the mature naïve B cell stage. BAFFR expression is reduced upon activation of naïve B cells and has low expression level on memory B cells. Binding of BAFF to BAFFR initiates a downstream signaling that is essential for the development of B cells [58,59,60,61,62,63]. Mice having either a deficiency in BAFF or BAFFR have a developmental block at the transitional stage, resulting in a complete loss of mature B cells. The TACI receptor is expressed by mature naïve B cells, memory B cells, and activated and marginal zone B cells, but not by GC B cells. Signaling through TACI leads to B cell differentiation, isotype switching, and immunoglobulin production. TACI-deficient mice have increased mature B cell numbers, suggesting that TACI attenuates the maturation of B cells [64,65,66]. BCMA was reported to be predominantly expressed by plasma cells and to provide survival signals for plasma cells in bone marrow [67]. BCMA-deficient mice display a normal phenotype and have normal B cell development and humoral immune response [68].

EAE studies using the BAFF and APRIL receptor knock-outs have identified an anti-inflammatory effect of BAFF and APRIL on B cells. In EAE induced with the MOG_35-55_ peptide in mice, deficiencies in BAFFR, which cause a developmental blockade early in B cell development at the transitional stage, increase disease severity [59,69,70]. This supports the theory that newly developed transitional B cells have anti-inflammatory properties in neuro-inflammation. However, BAFF and APRIL also have effects on B cells at later stages of maturation, which could impact neuro-inflammation [71,72,73]. Unlike BAFFR deficiency, BCMA-deficient mice have no overt defects in the development and homeostasis of B cell populations, although it has been reported that there are effects on antigen presentation and the maintenance of long-lived plasma cells [68,74]. We reported that BCMA-deficient (BCMA^−/−^) mice had significantly increased disease compared with their littermate control (BCMA^+/+^) mice. The increased disease in the BCMA^−/−^ mice was associated with a decrease in anti-inflammatory transitional and CD5^+^ B cells and an increase in inflammatory class-switched memory B cells and elevated IgG antibody responses to myelin antigen [75]. Mechanistically, we found that transitional B cells from BCMA^−/−^ mice secreted less anti-inflammatory IL-10 compared with BCMA^+/+^ transitional B cells. We also found that class-switched memory B cells from BCMA^−/−^ secreted higher levels of the inflammatory IL-6 in B cells compared with BCMA^+/+^ memory B cells. These data from BCMA^−/−^ mice suggest that BAFF and/or APRIL act downstream of the early developmental checkpoints of B cells and act to enhance anti-inflammatory B cell activity and provide a brake on the development of inflammatory B cell populations. TACI also functions at later stages of B cell development, and TACI-deficient mice also have no defects in B cell development; however, the specific role of TACI in EAE has not been reported.

In MS, elevated levels of BAFF are observed in the spinal fluid during relapses in patients. The elevation of BAFF during a relapse seems counterintuitive; however, this could be a physiological attempt to reduce inflammation during an acute attack. Interestingly, it was demonstrated that a soluble form of BCMA is elevated in the spinal fluid of MS patients and is positively correlated with intrathecal IgG production [76]. Therefore, it is plausible that soluble BCMA would counteract the elevated levels of BAFF in the CNS during a relapse and block the anti-inflammatory function BAFF has on B cells.

Overall, the clinical observations from the atacicept clinical trial and the experimental data from MS patients and EAE mice demonstrate that BAFF and APRIL play a critical role in maintaining a healthy balance of inflammatory and anti-inflammatory B cells and positively impact the disease outcome of MS patients. Therefore, research focusing on exploiting BAFF and APRIL signaling mechanisms may reveal new therapeutic strategies for MS.

## 5. Conclusions

B cells have dual functions in MS, having both inflammatory as well as anti-inflammatory functions in the disease. Clinical studies with MS therapies suggest that memory B cells are the major inflammatory B-cell subset that facilitate the initiation and perpetuation of CNS inflammation. Most MS treatments significantly decrease the memory B cells, thereby reducing the activation of pathogenic T cells, which ultimately reduces disease severity. Plasmablasts and plasma-cells-producing autoantibodies are another subset that could promote worsening of disease, though the ability of plasma cells and autoantibodies to significantly affect the course of MS is still a subject of debate among researchers. Contrarily, anti-inflammatory B cell subsets function to attenuate the progression and severity of disease. MS therapies that accelerate anti-inflammatory B cell development or IL-10 secretion have beneficial therapeutic effects in MS. Different strategies used to target B cells have achieved success in reducing disease severity, but an efficient, safe, and long-lasting therapy is still needed. Designing new B cell-targeting therapies that specifically eliminate inflammatory B cells but spare or expand anti-inflammatory B cells could be explored as an approach toward developing a safer and more effective therapy. Therefore, research that provides a sound knowledge of the diverse functions of B cell subsets in MS is needed for the development of future therapies for MS.

## Figures and Tables

**Figure 1 ijms-24-02336-f001:**
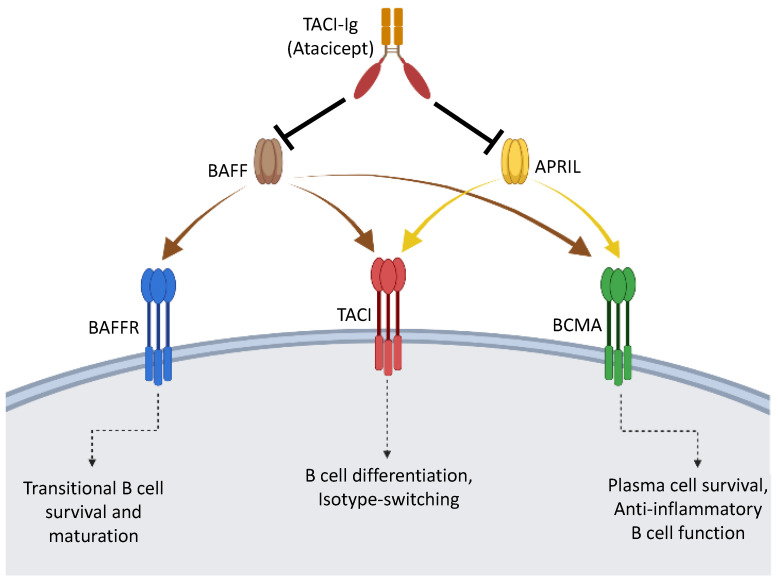
B cell activating factor (BAFF) and a proliferation-inducing ligand (APRIL) are cytokines that bind to BAFF receptor (BAFFR), transmembrane activator and calcium modulator and cyclophilin ligand interactor (TACI), and B cell maturation antigen (BCMA) expressed on the surface of B cell subsets. BAFF and APRIL signal through the receptors to affect B cell subset development, survival, and function. Recombinant TACI-Ig (atacicept) inhibits BAFF and APRIL function by blocking binding to their receptors.

**Figure 2 ijms-24-02336-f002:**
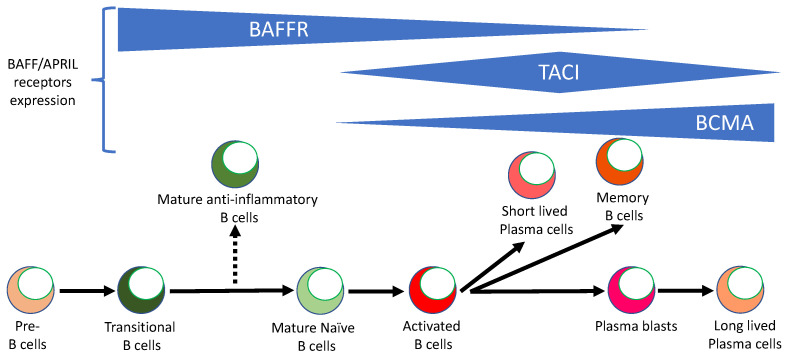
BAFFR, TACI, and BCMA are expressed at different stages of B cell development. After exiting the bone marrow, B cell precursors (pre-B cells) differentiate into immature transitional B cells that have anti-inflammatory functions. Transitional B cells then give rise to mature naïve and mature anti-inflammatory B cell subsets. Mature naïve B cells can become activated and further differentiate into short-lived plasma cells, memory B cells, and plasmablasts, which finally become long-lived plasma cells. All three BAFF and APRIL receptors are expressed at various levels on B cell subsets. BAFFR is highly expressed from the transitional B cells to the mature naïve B-cell stage. TACI is highly expressed from the mature naïve B cells to the activated and memory B cell stages. BCMA is most highly expressed by plasmablasts and short- and long-lived plasma cells.

**Table 1 ijms-24-02336-t001:** Inflammatory and regulatory B cell subsets in MS/EAE.

Inflammatory B Cells	Source	Functions
Memory B cells	MS patients	Antigen presentation
		Pro-inflammatory cytokine production
	EAE mice	Exacerbates EAE
		Produces IL-6
IgG + plasma cells	CNS Lesions in MS patients	Oligoclonal bands
		Auto-antibody production
	EAE mice	Facilitate CNS damage
		Increases disease severity
**Regulatory B Cells**		
Naïve B cells	MS patients	High levels of IL-10
Bregs/Transitional B cells	MS patients	Produce IL-10, IL-35, TGF-β
		Suppresses TNF production by monocytes.
	EAE mice	Produce IL-10
		Inhibits TH1 and TH17 cells
IgA + plasma cells	EAE mice	Attenuates EAE
		Produces IL-10

**Table 2 ijms-24-02336-t002:** Effects of MS therapies on B cells.

MS Therapies	Mechanism of Action
Anti-CD20	Removes all B cell subsets except pro-B cells and plasma cells
	Depletes IL-6 producing B cells
Interferon	Elevates serum BAFF levels
	Increases IL-10 producing transitional B cells
	Decreases memory B cells
Dimethyl fumarate	Reduces memory B cells
	Spares transitional B cells
	Decreases pro-inflammatory cytokines GM-CSF, IL-6 and TNF-α
Fingolimod	Blocks egression of inflammatory cells from secondary lymphoid tissues.
	Reduced circulating lymphocytes
	Reduced CNS infiltration of inflammatory cells
	Reduces memory B cells
	Increases number and CNS infiltration of regulatory B cells

## Data Availability

Not applicable.

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
