# Peer review of "Dual Role of B Cells in Multiple Sclerosis"

_ijms, 2023, doi:10.3390/ijms24032336_

Round 1

Reviewer 1 Report

I thank the authors for their extensive review of the role of B cells in multiple sclerosis.

I have a few comments for this review.

1. There is an error in the title of section 1 "1. ntroduction:", in addition, there are other errors in the text.

2. It seems to me that more attention should be paid to the novelty, significance, and most importantly, the purpose of this review.

3. It is necessary to add graphic material (figures, diagrams).

After these changes, it will be possible to re-consider the review and decide on its publication.

Author Response

We thank the review for the constructive comments for our article.  We have addressed your comments below:

  1. There is an error in the title of section 1 "1. ntroduction:", in addition, there are other errors in the text.  (We have thoroughly, proof-read this version of the manuscript)

2. It seems to me that more attention should be paid to the novelty, significance, and most importantly, the purpose of this review.

We have extensively revised this article and expanded on the experimental evidence that the BAFF and APRIL pathway is driving an anti-inflammatory B cell function in CNS autoimmunity

3. It is necessary to add graphic material (figures, diagrams).  We have added 2 figures to this article.

Reviewer 2 Report

Structure

This review reads like an enumeration of different studies without a clear interpretation of the available data or a clear story to follow. It is unclear why “4. B cells induce expansion of pathogenic T cells in periphery” and “5. B cells exacerbate neuronal damage in CNS” are different topics and not integrated within “3.1. Inflammatory subsets of B cells”, because these two paragraphs describe pathogenic roles of B cells. In addition, these two paragraphs do not conclude with an interpretation or why these characteristics of B cells are important within the review. A more structured approach would be to describe pro-inflammatory and anti-inflammatory (regulatory) B cells in MS and then immediately “6. Therapies that restrict inflammatory B cells and augment regulatory B cells in MS”. This way you can describe the B cell immune landscape in MS and how treatment changes these dynamics.

The part of the review about BAFF and APRIL is an interesting topic. However, this part reads like “another review within a review” without clear anchors in the sections above. Is there a way to integrate “7. BAFF and APRIL ensure B cell function” and “8. BAFF modulates non-B cell functions”? In “6. Therapies that restrict inflammatory B cells and augment regulatory B cells in MS” the effect on the levels of BAFF have already been illustrated for some treatment modalities, but without a detailed description about BAFF. Moreover, “BAFF modulates non-B cell functions” feels out-of-scope for this review.

In “6. Therapies that restrict inflammatory B cells and augment regulatory B cells in MS” several treatments are enumerated such as interferons, dimethyl fumarate, fingolimod. To me it is far-fetched to attribute to these treatments a mechanism of action that is primarily based on influencing B cells.

Language

There are a lot of writing and grammatical errors in this review; sometimes multiple in one sentence. This might reflect a lack of thoroughness and diligence (this is at least the feeling you get as a reader of this review).

It would be useful to use the term ‘regulatory’ in a consistent way. ‘Regulatory’ relates to the combination of inflammatory and anti-inflammatory effects, but is sometimes used as anti-inflammatory only.

References

Troughout this review various sentences/statements are not supported by their reference(s). An example is “In MS patients, autoantibodies, B cells and cytokines that support B cell function (BAFF, APRIL, IL-6, IL-21 and CXCL13) are found in CSF of MS patients during relapses [27]…”. The reference [27] does not provide any information about autoantibodies, B cells, APRIL or IL-21.

Also, sometimes a statement is made without any reference.

Tables

The tables do not provide a clear overview of the information presented. For example, in “Table 2. Effect of MS therapies on B cells.”, there are multiple ways to express a decrease in inflammatory lesions (“Reduces number of lesions”, “Reduced CNS inflammation”, “Decreases CNS inflammatory lesions”). These wordings make it very difficult to group treatments with a similar effect. It would have been a lot easier to understand if one and same wording was used (e.g., “Reduction inflammatory lesions”) or arrows to indicate a reduction or an increase.

NMOSD

Since this is a review about MS, it is often confusing when a study/reference about NMOSD is used. An example is “In addition, we and others have found that proteomic and transcriptomic markers provide strong evidence that RRMS is immunologically heterogeneous, and these markers may be used to indicate differences in treatment response and prognosis [6]”. This reference is about NMOSD and EAE, but states nothing about RRMS patients. In “6.1. Anti-CD20 therapies”, a whole part of the paragraph is dedicated to inebilizumab. However, inebilizumab is only approved for the treatment of AQP4+ NMOSD, a more B cell-driven autoimmune disorder in comparison to MS.

Lack of clinical background

In the review, there is often a lack of basic clinical knowledge. An example is “The criteria define MS as a patient with CIS whose brain MRI shows an earlier episode of damage in a different location along with a second active inflammation point in addition to the one that is causing the current symptoms.” This definition is not in agreement with the 2017 revised McDonald criteria. Another example is: “T cell targeted therapies”. There are no T cell targeted therapies for MS. …

Author Response

We thank the reviewer for the constructive comments. We have heavily edited the article to address the issue brought up by this reviewer, which we feel provides interpretation of the available data on the existence of both inflammatory and anti-inflammatory B cells in MS. 

1. We have expanded the section how BAFF and APRIL play drive an anti-inflammatory B cell function in MS and EAE. This section is required as it provides both clinical and experimental evidence that anti-inflammatory B cells do exist in MS.   

2. We have removed reference to NMO.

3. We have edited table 2 to focus on the effects of B cells.

4 .We also have thoroughly proof-read this version of the manuscript and have appropriately added references. 

5. We have streamlined the introduction to eliminate the lack of clinical background.

Round 2

Reviewer 1 Report

I thank the authors for their attentive attitude to the comments.

There is only one recommendation left: please replace figure 1, since it almost matches the figure from the application found in a Google search.

Author Response

Thanks again for the reviews. We have edited figure 1.